# A Single-Cell Atlas of an Early Mongolian Sheep Embryo

**DOI:** 10.3390/vetsci10090543

**Published:** 2023-08-28

**Authors:** Tingyi He, Wenrui Guo, Guang Yang, Hong Su, Aolei Dou, Lu Chen, Teng Ma, Jie Su, Moning Liu, Budeng Su, Wangmei Qi, Haijun Li, Wei Mao, Xiumei Wang, Xihe Li, Yanyan Yang, Yongli Song, Guifang Cao

**Affiliations:** 1Inner Mongolia Key Laboratory of Basic Veterinary Medicine, Key Laboratory of Animal Embryo, and Development Engineering Autonomous Region, Inner Mongolia Agricultural University, Hohhot 010018, China; yier.goal@163.com (T.H.); gwrui@163.com (W.G.); hongsu1995@126.com (H.S.); douaolei@126.com (A.D.); chenlu736@163.com (L.C.); mateng20230601@126.com (T.M.); momo4237@126.com (M.L.); subudaimau@163.com (B.S.); qiwangmei@imau.edu.cn (W.Q.); navy1973@163.com (H.L.); maowei2014@imau.edu.cn (W.M.); wangxiumei62@163.com (X.W.); 2Institute of Animal Husbandry, Inner Mongolia Academy of Agricultural and Animal Husbandry Sciences, Huhhot 010031, China; 3The State Key Laboratory of Reproductive Regulation and Breeding of Grassland Livestock, College of Life Sciences, Inner Mongolia University, Hohhot 010020, China; yangguangimu@163.com (G.Y.); lixh@imu.edu.cn (X.L.); 4Research Center for Animal Genetic Resources of Mongolia Plateau, College of Life Science, Inner Mongolia University, Hohhot 010020, China; 5Department of Medical Neurobiology, Inner Mongolia Medical University, Huhhot 010030, China; sujie0429@126.com

**Keywords:** Mongolian sheep, embryo, scRNA-seq, Hippo signaling pathway

## Abstract

**Simple Summary:**

This study presents the first comprehensive single-cell transcriptomic characterization at E16 in Ujumqin sheep and Hulunbuir short-tailed sheep (the day of mating was defined as day 0, and embryo samples were taken on the 16th day). TBXT mutations are related to tailless or short-tailed phenotypes in vertebrates. In our previous study, we detected the TBXT expression level at a different embryo stage in sheep and found that TBXT exhibited the highest expression at E16. We believe that TBXT plays an important role in E16; thus, we chose E16 as the focus of this study. This comprehensive single-cell map reveals previously unrecognized signaling pathways that will improve our understanding of the mechanism of short-tailed sheep formation.

**Abstract:**

Cell types have been established during organogenesis based on early mouse embryos. However, our understanding of cell types and molecular mechanisms in the early embryo development of Mongolian sheep has been hampered. This study presents the first comprehensive single-cell transcriptomic characterization at E16 in Ujumqin sheep and Hulunbuir short-tailed sheep. Thirteen major cell types were identified at E16 in Ujumqin sheep, and eight major cell types were identified at E16 in Hulunbuir short-tailed sheep. Function enrichment analysis showed that several pathways were significantly enriched in the TGF-beta signaling pathway, the Hippo signaling pathway, the platelet activation pathway, the riboflavin metabolism pathway, the Wnt signaling pathway, regulation of the actin cytoskeleton, and the insulin signaling pathway in the notochord cluster. Glutathione metabolism, glyoxylate, and dicarboxylate metabolism, the citrate cycle, thyroid hormone synthesis, pyruvate metabolism, cysteine and methionine metabolism, thermogenesis, and the VEGF signaling pathway were significantly enriched in the spinal cord cluster. Steroid biosynthesis, riboflavin metabolism, the cell cycle, the Hippo signaling pathway, the Hedgehog signaling pathway, the FoxO signaling pathway, the JAK-STAT signaling pathway, and the Wnt signaling pathway were significantly enriched in the paraxial mesoderm cluster. The notochord cluster, spinal cord cluster, and paraxial mesoderm cluster were found to be highly associated with tail development. Pseudo-time analysis demonstrated that the mesenchyme can translate to the notochord in Ujumqin sheep. Molecular assays revealed that the Hippo signaling pathway was enriched in Ujumqin sheep. This comprehensive single-cell map revealed previously unrecognized signaling pathways that will further our understanding of the mechanism of short-tailed sheep formation.

## 1. Introduction

Ujumqin sheep and Hulunbuir short-tailed sheep (Hulunbuir sheep) are famous and profitable sheep breeds in the Inner Mongolia Autonomous Region, China. The Ujumqin sheep have a fat tail and multi-vertebral characteristics [1], while Hulunbuir short-tailed sheep have smaller tails and less fat accumulation. Adipose tissue is very important for tail development in domestic animals [2]. Due to the short-tailed phenotype of the Hulunbuir short-tailed sheep, artificial and natural selection were applied to increase meat production [3]. T-box transcription factor (TBXT) encodes the Brachyury protein and is involved in mesoderm formation and differentiation [4]. Previous studies showed that TBXT mutations are related to tailless or short-tailed phenotypes in vertebrates. Missense mutations of c.334G>T and c.334G>C are associated with the tailless characteristic. Brachyury functions as a transcriptional activator, mesoendoderm marker, and key development regulator. The gene regulatory loop of Brachyury, in addition to Sox17 and Foxa2, directly regulates stem cell lineage commitment during streak formation. Brachyury and Brachyury-expressing cells can differentiate into mesodermal and definitive endodermal lineages and play diverse and indispensable roles in early mouse embryo development. The target genes of Brachyury include Dusp6, Mesp2, Hoxd8, Cdx2, Irx3, Lefty, Msgn1, Dhrs3, and sp5 in zebrafish, *X. tropicalis*, and mice. Brachyury-bound genes include Fgf8, Sox17, Foxa2, Tbx3, Wnt9b, and Rest [5]. However, some short-tailed phenotypes are not caused by Brachyury mutation. For instance, in domestic cats, the substitution of c.5T>C in the somite segmentation-related gene HES7 is associated with the tailless characteristic. The tailed trait in cats could not be explained by either HES7 or the Manx-related T-box, suggesting at least three independent events in the evolution of domestic cats, giving rise to the short-tailed trait [6]. The Gyeongju Donggyeong dog is iconic in Korea. A unique feature of the Donggyeong dog is that it is either short-tailed or tailless [7]. T-box mutation is present in most, but not all, short-tailed dog breeds [8]. ANKRD11 and ACVR2B may contribute to reduced tail length, and the T gene has a non-synonymous variant in the coding regions. However, the CPG island variant of the SFRP2 gene can be a candidate causal variant for the Donggyeongo dog tailless trait [9]. The short-tail genetic mechanism of the Donggyeongo dog has yet to be fully elucidated. The sheep embryo cell cluster also remains unclear.

Single-cell RNA sequencing (scRNA-seq) techniques have enabled important advances in tissue development and disease diagnosis in the past decade [10]. This technology could illustrate the heterogeneity of biological samples [11,12]. Common methods, including functional transplantation assays, can be used for cell lineage tracing and to reveal the cellular roadmap from tissue [13,14,15]. scRNA-seq is a powerful approach used to help us understand cell development trajectories in humans and mammals [16]. Mammalian organogenesis is an astonishing process. During embryo formation, the transcriptional dynamics of mouse organogenesis have been reported, including for E9.5 and E13.5 mouse embryos [17]. scRNA-seq plays an important role in understanding tail development during embryogenesis. The expression level of FGF/WNT signaling genes and tailbuds declined in the differentiated cells [18]. During somite formation, WNT, Notch, and FGF signaling waves traveled from the tail bud towards the differentiation front [19]. Furthermore, spatial transcriptomics analysis of somitogenesis in gastruloids unraveled the complex processes involved in the WNT, Notch, and FGF signaling pathways that regulate embryogenesis [20]. However, these cell populations and molecular mechanisms have not been elucidated in early embryos of Mongolian sheep. In this study, for the first time, we used scRNA-seq to establish the cell expression profile in early embryos of Ujumqin sheep and Hulunbuir short-tailed sheep. The cell populations, marker gene identification, and signaling pathway were analyzed. TBXT mutations are related to tailless or short-tailed phenotypes in vertebrates. We found that TBXT is most highly expressed at E16. However, it has yet to be determined which cells and signaling pathways are associated with short sheep tails at E16. Scientists have studied gene expression and cell types in early embryos in mice and humans. However, the cell type of sheep E16 remains unclear. Our results may provide a foundation for further research on early embryo development in sheep.

## 2. Methods

### 2.1. Experiment Animals

Sheep are very important to the economy in inner Mongolia, China. After identifying the short-tailed phenotype in Hulunbuir sheep, we chose to focus on this breed in our research; Ujumqin sheep with the long-tailed phenotype were selected as the control group. Healthy Ujumqin sheep (24 months, 50 kg, autumn) were obtained from a Ujumqin sheep breeding farm in Xilingol League, Inner Mongolia Autonomous Region, China. Healthy Hulunbuir sheep (24 months, 50 kg, autumn) were sourced from Inner Mongolia Mengyuan mutton sheep breeding industry Co., Ltd., Baotou, China.

### 2.2. Embryo Sample Collection and Preparation of Single-Cell Embryo Cell Suspension

Twenty Ujumqin and forty Hulunbuir healthy sheep, all aged 24 months, were treated with progesterone vaginal sponge embolization for 12 days. PMSG (330 IU/sheep) was intramuscularly injected prior to thrombectomy; estrous ewes were mated after 36 h. Embryos were collected on day 16 of pregnancy via surgical operation. The embryos were washed three times using cold PBS. Extraembryonic tissue was carefully removed using sterile pointed forceps under a microscope and then immediately transferred to tissue storage solution (Miltenyi, Bergisch Gladbach, Germany, 130-100-008) with 3% penicillin–streptomycin at room temperature. A single-cell suspension of early sheep embryos was prepared as described in [21]. The cell number and viability were measured using Countess 3 (Thermo Fisher Scientific, Waltham, MA, USA). Cell viability > 85% was explored as scRNA-seq.

### 2.3. Embryos Single Cell RNA-Seq Performances and Sequencing

The cells were held on ice before being loaded on a GemCode single-cell platform (10×). Chromium Single Cell 3′ v3 libraries were sequenced using a Novaseq 6000 sequencer; the qualified cDNA library was sequenced by Analytical Biosciences (Beijing, China).

### 2.4. Mapping and Clustering Analysis

Raw Illumina data were demultiplexed and processed using Cell Ranger (10 × Genomics version Cell Ranger 5.0.0). Ovis reference transcriptome (rambouillet_v1.0) provided by 10 × genomics was used for mapping in analytical biosciences in China. Seurat version 3.2.0 was used for filtering and subsequent clustering [22]. Low-quality cells were removed via quality control filtering [23]. Additionally, the mitochondrial percentage per cell was less than 10%. UAMP (uniform manifold approximation and projection) was performed using the highly variable genes to visualize the single-cell clustering [24]. A likelihood ratio test for single-cell gene expression was used to identify marker genes for each population [25]. In this study, single-cell consensus clustering analysis was used to validate the robustness of some clusters [26]. Cell cycle analysis was carried out in Seurat using a list of cell cycle genes from the Regev laboratory [27] and Paget paper for analysis methods [28]. Single-cell RNA-sequencing data have been submitted to the GEO repository using the access code GSE185233.

### 2.5. High-Density Plots

High-density plots were generated using the MATLAB function with gene cell counts and cluster labels as the input and visualized [29].

### 2.6. Pseudo-Time Analysis

Monocle version 2.10.1 was used on cells filtered from Seurat to infer differentiation trajectories [30]. An expression threshold of 0.1 was applied. The highly variable genes identified from Seurat were used as the ordering filter [31], and then the trajectory was obtained. To identify pseudo-time-dependent gene expression changes, we applied the single-cell energy path (scEpath; Version 1; MATLAB Version 9.5) [32].

### 2.7. RNA Velocity

RNA velocities were calculated based on spliced and unspliced transcript reads, as previously reported [33].

### 2.8. qRT-PCR Analysis

RNA was isolated from whole Ujumqin and Hulunbuir sheep at E16 using RNAfast200 (220011, Shanghai, China) according to the manufacturer’s instructions. Reverse transcription was performed using a PrimeScript RT Reagent Kit with a gDNA Eraser (RR047A Takara, Kusatsu, Japan). Real-time PCR was performed using the ChamQ Universal SYBR qPCR master mix on a CFX96 PCR system (Bio-Rad, Hercules, CA, USA). The housekeeping gene Gapdh was used as an internal control, and gene expression was calculated based on the ΔΔCT method. The primer sequences used are shown in Appendix A.

### 2.9. Statistical Analysis

Unless otherwise specified, all analyses were performed in at least triplicate. Statistical analysis was performed using one-way ANOVA and Student’s *t*-test (* *p* < 0.05, ** *p* < 0.01, and *** *p* < 0.001). All data are reported as mean ± SD. The means and standard deviations from at least three independent experiments are represented in all graphs.

## 3. Results

### 3.1. Cell Type Identification and Cluster Analysis at E16 in Ujumqin and Hulunbuir Sheep

To begin to address the cell population and markers of early embryos in Ujumqin and Hulunbuir sheep, we collected Ujumqin and Hulunbuir sheep embryos at E16. Following dissociation, embryos were pooled and processed on 10 x genomics (Figure 1A). We obtained the single-cell suspension with an 88.1% cell viability and concentration of 1.43 × 10^6^ cells/mL in Ujumqin E16 and with a 92.22% cell viability and concentration of 1.24 × 10^6^ cells/mL in Hulunbuir E16 (Appendix A). We processed a total of ~27,768 cells (Ujumqin sheep: *n* = ~17,059 cells; Hulunbuir sheep: *n* = ~10,709 cells). Next, we used Seurat’s Anchoring algorithm [34] to integrate the Ujumqin and Hulunbuir sheep datasets and visualized them using Uniform Manifold Approximation and Projection (UMAP) (Appendix A), and performed quality control analysis on individual libraries (Appendix A). The number of different cell types was determined in Ujumqin sheep E16 (Appendix A) and in Hulunbuir sheep E16 (Appendix A).

To further study the E16 cell diversity of Ujumqin and Hulunbuir sheep, a variety of biomarkers were explored for the identification of cell clusters (Appendix A). We subclustered the cell diversity and identified thirteen distinct cell clusters, including the blood progenitor, cardiomyocytes, caudal neuroectoderm, endothelium, erythroid, ExE endoderm, gut, hematoendothelial progenitors, mesenchyme, notochord, paraxial mesoderm, somite, and spinal cord in Ujumqin sheep (Figure 1B). We also identified eight distinct cell clusters, including the endothelium, gut, hematoendothelial progenitors, notochord, paraxial mesoderm, primitive/cardiomyocytes, somite, and spinal cord in Hulunbuir sheep (Figure 1C). Interestingly, when split by condition, mesenchymes appear to be specific to Ujumqin sheep, suggesting that these communities are altered in Ujumqin sheep compared with Hulunbuir sheep. It is worth noting that mesenchymes are associated with the formation of the tail in Ujumqin sheep. Next, we determined the top 20 genes expressed in every cell cluster in Ujumqin sheep. The expression levels of PTN, IGFBP2, and ID3 were higher in the spinal cord. COL1A1, COL1A2, IGF2, APOE, and FKBP11 were highly expressed in the somite. CRABP1, ID3, COL2A1, FST, FOS, and IGF2 were highly expressed in the notochord. ID3, IGF2, IGFBP2, and TOP2A were highly expressed in the paraxial mesoderm in Ujumqin sheep (Figure 1D). We detected the top 20 genes expressed in every cell cluster in Hulunbuir sheep. The expression levels of IGFBP5, IGF2, IGFBP7, and POSTN were higher in paraxial mesoderm in Hulunbuir sheep. Compared with the paraxial mesoderm in Ujumqin sheep, we found that IGFBP2 was more highly expressed in Ujumqin sheep. However, IGFBP5 and IGFBP7 exhibited higher expression in Hulunbuir sheep. Thus, different IGFBP family members play a significant role in the formation of the tail in these sheep. COL1A1, COL1A2, APOE, and FKBP11 were highly expressed in the somite in Hulunbuir sheep. PTN, IGFBP2, and PAX6 were highly expressed in the spinal cord in Hulunbuir sheep. However, whilst PTN and IGFBP2 were expressed in two samples, PAX6 was only highly expressed in the spinal cord in Hulunbuir sheep. This shows that PAX6 plays an important role in tail formation in Hulunbuir sheep. IGF2, FGF8, and WNT5B were highly expressed in the notochord in Hulunbuir sheep. However, whilst IGF2 was expressed in two samples, FGF8 and WNT5B were only highly expressed in the spinal cord in Hulunbuir sheep. This indicates that FGF8 and WNT5B play an important role in tail formation in Hulunbuir sheep (Figure 1E). We also observed COL2A1, FOS, MEIS2, and NR2F2 expression in two samples. Interestingly, we found that these genes were specially expressed in the notochord in Ujumqin sheep. However, the expression of these genes was limited in the notochord in Hulunbuir sheep. These results demonstrated that COL2A1, FOS, MEIS2, and NR2F2 could participate in tail formation and regulate the tail morphology in Ujumqin sheep (Figure 1F,G). Collectively, our data suggest that the cell type was first identified at E16 in Ujumqin and Hulunbuir sheep, and some key factors regulating tail morphology were confirmed at E16 in two samples.

### 3.2. Functional Enrichment Analysis of E16 in Ujumqin and Hulunbuir Sheep

The notochord, spinal, and paraxial mesoderm cell clusters play a key role in tail formation in Ujumqin and Hulunbuir sheep. KEGG analyses were performed to investigate the different functions of the notochord, spinal, and paraxial mesoderm cells. We observed that the TGF-beta signaling pathway, Hippo signaling pathway, platelet activation pathway, riboflavin metabolism pathway, Wnt signaling pathway, regulation of actin cytoskeleton, and insulin signaling pathway were enriched in the notochord of Ujumqin sheep (Figure 2A). However, our data showed that the Hippo signaling pathway could regulate tail morphology in sheep. Glutathione metabolism, glyoxylate, and dicarboxylate metabolism, the citrate cycle (TCA cycle), thyroid hormone synthesis, pyruvate metabolism, cysteine and methionine metabolism, thermogenesis, and VEGF signaling pathway were enriched in the spinal cord of Ujumqin sheep (Figure 2B). Steroid biosynthesis, riboflavin metabolism, the cell cycle, Hippo signaling pathway, Hedgehog signaling pathway, FoxO signaling pathway, JAK-STAT signaling pathway, and Wnt signaling pathway were enriched in the paraxial mesoderm in Ujumqin sheep (Figure 2C). Our data showed that the Hippo signaling pathway plays an important role in spinal cord and paraxial mesoderm development. To further study the gene expression profile in notochord spinal cord and paraxial mesoderm cell clusters, we observed that IGFBP5^+^ and FST^+^ expression is mainly located in the notochord in Ujumqin sheep; however, IGFBP5^+^ expression is mainly in located in the somite and FST^+^ expression mainly occurs in the spinal cord in Hulunbuir sheep (Figure 2D). These data show that IGFBP5 and FST play different roles in tail development. We also found that AHSG^+^ and AFP^+^ expression is mainly located in the spinal cord in Ujumqin sheep but was not detected in the cell cluster in Hulunbuir sheep (Figure 2E). These results suggest that AHSG and AFP could be candidate genes that allow Ujumqin and Hulunbuir sheep to be distinguished from early embryos and lambs. This finding has significant economic and breeding value. HOXB9 expression is mainly located in the paraxial mesoderm, notochord, and spinal cord in Hulunbuir sheep. Together, these results reveal the HOXB9 expression pattern in early sheep embryos (Figure 2F). Selected DEGs were confirmed using qRT-PCR analysis, and the AHSG, AFP, FST, and VIM (Vimentin) expression levels decreased significantly while the HOXB9 expression level increased in Hulunbuir sheep (Figure 2G). These findings are in line with scRNA-seq data.

### 3.3. Different Expression Gene Analysis in Every Cell Cluster at E16 in Ujumqin and Hulunbuir Sheep

To gain insights into the more detailed information for different expression genes in every cluster at E16 in Ujumqin and Hulunbuir sheep, we compared key gene expression data in two samples. We observed that ELOV6 (Elongase of very long-chain fatty acid 6) is more highly expressed in Hulunbuir sheep compared with Ujumqin sheep, especially in the endothelium, notochord, paraxial mesoderm, somite, and spinal cord. ELOV6 is a speed-limiting enzyme for long-chain fatty acid elongation reaction. When ELOV6 is overexpressed, the synthesis of long-chain fatty acid is limited. This result reveals that ELOV6 could regulate tail fat morphology. We also found that ACLY (ATP-citrate Lyase) and MECR (Mitochondrial Trans-2-Enoyl-CoA Reductase) were more highly expressed in the notochord, paraxial mesoderm, somite, and spinal cord in Hulunbuir sheep (Figure 3A). These results reveal that ACLY and MECR could influence the tail fat morphology by regulating fatty acid biosynthesis. PPP1CC (Protein Phosphatase 1 Catalytic Subunit Gamma) regulates many cellular processes, including cell division. PPP1CC was highly expressed in the notochord, paraxial mesoderm, somite, and spinal cord in Ujumqin sheep (Figure 3B). This result showed that PPP1CC could regulate the tail morphology by influencing cell division. We also detected that REN (extracellular matrix serine protease), ITGA8 (Integrin Submit Alpha8), HOXC10, and HOXC11 were more highly expressed in Hulunbuir sheep compared with Ujumqin sheep. Renin is a part of the renin–angiotensin–aldosterone system involved in the regulation of blood pressure and electrolyte balance. However, its function in sheep has yet to be determined. ITGA8 regulated numerous cellular processes, including cell adhesion. HOX10, a homeobox family of genes, provides cells with specific positional identities on the anterior–posterior axis. HOX11 regulates mesodermal commitment. These data demonstrate that Ujumqin and Hulunbuir sheep can be identified by confirming the gene expression level (Figure 3C,D). Next, we compared VIM (Vimentin), LGALS3, ANXA1 (Annexin A1), and RARRES1 (Retinoic Acid Receptor Responder 1). VIM, LGALS3, ANXA1, and RARRES1 were highly expressed in Ujumqin sheep. However, their functions in sheep have yet to be determined. We believe that they could regulate embryo development; they could also be candidate genes that may allow us to identify Ujumqin and Hulunbuir sheep by assessing their gene expression level (Appendix A). Next, we detected the expression of fat development-related genes. We found that PAX3 and T (Brachyury) were more highly expressed in Hulunbuir sheep compared with Ujumqin sheep (Appendix A). We detected the CDX2 expression level via qPCR in Hulunbuir and Ujumqin sheep (Appendix A). The results showed that the CDX2 expression level was higher at E16 in Hulunbuir sheep compared with Ujumqin sheep, and the CDX2 expression pattern is similar to the T expression pattern. CDX2 is mainly located in the spinal cord in Ujumqin sheep; however, CDX2 is mainly located in the notochord in Hulunbuir sheep. The CDX2 expression level may be a candidate gene with which to identify Hulunbuir and Ujumqin sheep.

### 3.4. Sc-RNAseq Analysis of Notochord, Paraxial Mesoderm, Somite, and Spinal Cord Cell Characteristics in Hulunbuir and Ujumqin Sheep

To gain insights into the differences in the molecular mechanisms of Hulunbuir and Ujumqin sheep, we compared the pathway change in the notochord, paraxial mesoderm, somite, and spinal cord in Hulunbuir and Ujumqin sheep. We observed that the Hippo signaling pathway and TGF-β signaling pathway changed in two samples. APOE is more highly expressed in the notochord in Ujumqin sheep and is related to fat development and pyruvate metabolism. Meanwhile, COL3A1, COL1A2, and COL1A1 were highly expressed in the somite and notochord in Ujumqin sheep. These proteins are related to protein digestion and absorption. This could explain why the morphology of the Uqumqin sheep includes a big tail (Figure 4A). We also noted high AFP expression in the spinal cord in Hulubuir sheep. This is in line with our data (Figure 4B). We also detected the YAP1, TAZ, ACTB, MYC, and AFP expression levels via qPCR in Hulubuir and Ujumqin sheep. The results are in line with scRNAseq data (Appendix A). The same results were observed in the notochord in Ujumqin and Hulubuir sheep (Figure 4C). At the same time, we detected the expression level of HOXB9, ACLY, FASN, and HOCX12 via qPCR in two samples (Appendix A). The result showed that the HOXB9 expression level was higher in Hulunbuir sheep, and the ACLY and HOXC12 expression levels were lower in Hulunbuir sheep. We also detected the BMP2, ACSL1, FABP4, PRKACA, ACACA, and ACSL1 expression levels via UMAP in Hulunbuir and Ujumqin sheep. The BMP2, ACSL1, and FABP4 expression levels were almost identical in Hulunbuir and Ujumqin sheep, and the ACACA expression level was higher in Hulunbuir sheep. BMP2, ACSL1, and FABP4 were related to fat development; thus, their higher expression could be related to big tail morphology (Appendix A). We also detected the HOXC12 and TNNC1 expression levels in Hulunbuir and Ujumqin sheep (Appendix A). The data revealed that the HOXC12 and TNNC1 expression levels were increased in Hulunbuir and Ujumqin sheep.

### 3.5. RNA Velocity and Gene Expression Analyses Support the Mesenchyme–Notochord Cell Transformation Theory in Ujumqin Sheep

This trajectory enabled the reconstruction of a major path, defined as H-PATH 1, in which the mesenchyme was placed at the beginning and notochord cells were placed at the trajectory terminus (Figure 5A). This suggests that a transition from mesenchyme to notochord cells is possible in Ujumqin sheep at E16. We also detected a mesenchyme–spinal cord cell transformation trajectory (Figure 5B). The results showed that a transition from the mesenchyme to the spinal cord is impossible in Ujumqin sheep at E16. After that, we focused on pseudo-time-dependent transcription factors (TFs) in the notochord, including AHSP, NENF, TMEM120A, TMEM14C, ALAS2, etc. (Figure 5C). We identified pseudo-time-dependent TFs that could be important for the development of notochord cells. We also focused on pseudo-time-dependent transcription factors (TFs) in the spinal cord, including IGFBP3, NES, FGFR1, HOXB9, KLHL24, PAX3, TOP2A, UPP1, etc. (Figure 5D). Furthermore, we identified pseudo-time-dependent TFs that could be important for the development of spinal cord cells. Together, our data revealed a putative mesenchyme–notochord cell transformation trajectory.

Next, we detected the cell cycle in Hulunbuir and Ujumqin sheep. Cell cycle state analysis showed that ~39% of spinal cord cells were in the G1 phase, while ~37% and ~24% of spinal cord cells were, respectively, in the S and G2/M phases in Ujumqin sheep. Approximately 25% of notochord cells were in the G1 phase, while ~35% and ~40% of notochord cells were, respectively, in the S and G2/M phases in Ujumqin sheep. Approximately 36% of somite cells were in the G1 phase, while ~30% and ~34% of somite cells were, respectively, in the S and G2/M phases in Ujumqin sheep. Approximately 39% of paraxial mesoderm cells were in the G1 phase, while ~29% and ~32% of paraxial mesoderm cells were, respectively, in the S and G2/M phases in Ujumqin sheep (Appendix A). Approximately 35% of spinal cord cells were in the G1 phase, while ~44% and ~21% of spinal cord cells were, respectively, in the S and G2/M phases in Hulunbuir sheep. Approximately 34% of notochord cells were in the G1 phase, while ~35% and ~41% of notochord cells were, respectively, in the S and G2/M phases in Hulunbuir sheep. Approximately 29% of somite cells were in the G1 phase, while ~47% and ~24% of somite cells were, respectively, in the S and G2/M phases in Ujumqin sheep. Approximately 37% of paraxial mesoderm cells were in the G1 phase, while ~28% and ~25% of paraxial mesoderm cells were, respectively, in the S and G2/M phases in Hulunbuir sheep (Appendix A). Collectively, the proliferation of spinal cord cells, notochord cells, somite cells, and paraxial mesoderm cells was faster in Ujumqin sheep compared with Hulunbuir sheep. These results suggest that the big tail morphology of Ujumqin sheep is related to faster cell proliferation compared with Hulunbuir sheep.

### 3.6. Cell–Cell Communication Analysis in Ujumqin and Hulunbuir Sheep

We performed cell–cell communication via cellphone DB. Interacting pairs were selected with a significance of *p* < 0.05. We found that of 486 somite/paraxial mesoderm and somite/mesenchymal communications, 487 were stronger in Ujumqin sheep (Figure 6A). This suggests that somite/mesenchymal interaction plays a key role in Ujumqin sheep during E16 development. IGF2, IGF2, and IGF1R have a significant role in cell–cell communication in Ujumqin sheep. We also found that cell–cell communication was generally stronger in Hulunbuir sheep (Figure 6B). MDK-SORL1 plays a key role in cell–cell communication in Hulunbuir sheep. Taken together, our data demonstrate that key genes play a significant role in cell–cell communication in Ujumqin and Hulunbuir sheep.

## 4. Discussion

Collectively, our results showed that the mesenchyme, paraxial mesoderm, and caudal neuroectoderm were more significant cell clusters in Ujumqin sheep than in Hulunbuir sheep. Epithelial-to-mesenchymal transition (EMT) appears to play a particularly vital role in the formation of tissue and organs [35]. Thus, mesenchymes have important roles in the formation of the tail in Ujumqin sheep. Previous research indicates that the platelet activation pathway plays an important role in fat deposition in the tails of sheep, and PDGFD was identified as a likely causal gene associated with tail morphology [36]. The Wnt signaling pathway and FGF signaling pathway also regulate tail morphology [20]. The TGF-beta signaling pathway plays a role in mouse tail formation [37]. Our results are in line with these studies. Previous research indicates that HOXB9 expression decreased in epiblast cells. However, HOXB9 expression increased from gastrulation in mice and bovines [38]. We also detected that HOXB9 expression is mainly located in the paraxial mesoderm in Ujumqin sheep; previous research showed that Pax3 overexpression leads to multiple defects, including aberrant myogenesis in the developing somites in mice [39]. The Pax3′ function remains unexplored in sheep embryo development. Our data showed that Pax3 was more highly expressed in the somite, spinal cord, and notochord in Hulunbuir sheep. This could result in aberrant somite, spinal cord, and notochord development, influencing tail morphology. Brachyury (T) plays a role in mesoderm formation in vertebrates [40]. In addition, the tail length of T-mutation mice was less than 1 cm to half of the normal length [41]. Thus, we detected T gene expression via UMAP in two samples. The result showed that the T gene expression level was higher in Hulunbuir sheep compared with Ujumqin sheep (Appendix A). Given that the Hulunbuir sheep is characterized by its short tail, its T gene expression level should be lower compared with that of the Ujumqin sheep. Interestingly, the T gene expression level was higher at E16 in Hulunbuir sheep compared with Ujumqin sheep. T may participate in notochord development. Previous research shows that CDX2 plays a key role in embryonic development [42]. The Hippo–YAP signaling pathway is an evolutionarily conserved signaling cascade and plays an essential role in controlling cell proliferation and development [43]. We note that YAP1, as a major transcription co-activator of the Hippo pathway, is essential for accomplishing the Hippo pathway function. Function analysis of genes related to Hippo signaling revealed that YAP1, WWTR1, STK3, STK4, and LATS1 were upregulated in the spinal cord in Hulubuir sheep. This suggests that Hippo signaling is inhibited in the spinal cord in Hulubuir sheep. Interestingly, a previous report showed that Yap positivity was associated with higher AFP levels [44]. To determine whether the mesenchyme can transform and assume a notochord cell fate in Ujumqin sheep, we inferred a putative mesenchyme–notochord cell transformation trajectory in pseudo-time using Monocle2 [30,45]. The cell cycle state analysis can be found in [27].

The quality of life and well-being of infants are impacted by successful pregnancy in humans. Thus, a thorough understanding of the developmental events that occur between conception and delivery is needed. Pregnant sheep can be used as a model for human pregnancy; thus, understanding early embryo development in sheep is an important research area [46]. A previous study showed that the morula and blastocyst development states are significantly different in the sheep proteome; 667 proteins were identified in the sheep embryo proteome [47]. In addition, DNMT1 maintains the methylation profile of genes during cell division; DNMT1 is associated with post-implantation mortality in sheep [48]. FGF2 is a member of the FGF family that mediates trophoblast activity and regulates embryonic development in various species [49]. This result is in line with our data. We found that FGF1/FGF1R and FGF2/FGF2R play important roles in E16 development in Mongolian sheep.

Previous research has outlined that the sheep gene expression atlas dataset expands on the RNA-seq datasets, including liver, spleen, ovary, testes, kidney, muscle, thymus, left ventricle, etc., for adult tissue. Early development (blastocysts), early development (day 23), early development (day 35), early development (day 100), maternal (days 23, 35, and 100), and COL1A1, COL1A2, and COL1A3 are related to embryo development [50]. These data are also in line with our data. This study provides annotation of the current reference genome for sheep and presents a model transcriptome for ruminants at multiple stages of sheep development. However, no report on the study of E16 in sheep has existed until this point. Our research fills this gap.

High embryonic mortality in ruminants necessitates increased knowledge of early embryo development. Failures in early embryonic development and placenta formation could be the result of abnormal mesoderm differentiation. Two T-box genes (Brachyury and Eomesodermin) control the gastrulation process. Whole-mount in situ hybridization is a precise method that compares the expression pattern between embryos. A previous report showed that Brachyury (T) is most highly expressed at E16 [51]. Brachyury (T) expression is also related to tail morphology. Thus, we chose E16 as a study subject in sheep. Interestingly, we found that Brachyury (T) expression is higher in Hulunbuir short-tail sheep compared with Ujumqin sheep, indicating a big tail morphology. This finding suggests that Brachyury (T) could take part in the gastrulation process rather than regulating tail morphology at this stage. Brachyury (T) plays different roles at different development stages in sheep’s early embryo development. Thus, more relational research needs to be carried out in early embryo development in sheep in the future.

Here, we report the Mongolian early sheep embryo (E16) transcriptional cell atlas; Mongolian sheep breeds include Ujumqin sheep, Hulunbuir sheep, etc. The Ujumqin sheep is characterized by its fat tail, while the Hulunbuir sheep have a smaller tail. We reported that the Ujumqin sheep has 13 cell clusters, and the Hulunbuir sheep has eight cell clusters, and many differential expression genes and signaling pathways were identified at E16 in the Ujumqin and Hulunbuir sheep. This study opens up an entirely new line of research in Mongolian sheep early embryo development and highlights the key factors regulating tail morphology.

Sheep embryogenesis is not well understood. In this study, we reveal the molecular and cell development landscape of E16 in Ujumqin and Hulunbuir sheep. These data provide vital insight into E16 development and uncover cell identification in Ujumqin and Hulunbuir sheep.

## 5. Conclusions

This study presented the first comprehensive single-cell transcriptomic characterization at E16 in Ujumqin sheep and Hulunbuir short-tailed sheep development. Thirteen major cell types were identified at E16 in Ujumqin sheep, and eight major cell types were identified at E16 in Hulunbuir short-tailed sheep. The notochord, spinal, and paraxial mesoderm cell clusters play an important role in tail formation in Ujumqin and Hulunbuir short-tailed sheep. Our data show that the Hippo signaling pathway could regulate the tail morphology of sheep in the notochord and paraxial mesoderm cell clusters. We also found that AHSG and AFP expression is mainly located in the spinal cord in Ujumqin sheep, while it was not detected in the cell cluster in Hulunbuir sheep. These results show that AHSG and AFP could be candidate genes for distinguishing Ujumqin and Hulunbuir sheep from early embryos and lambs. PPP1CC (Protein Phosphatase 1 Catalytic Subunit Gamma) regulates many cellular processes, including cell division. PPP1CC was highly expressed in the notochord, paraxial mesoderm, somite, and spinal cord in Ujumqin. This result showed that PPP1CC could regulate the tail morphology by influencing cell division. POE is related to fat development and pyruvate metabolism. APOE is highly expressed in the notochord in Ujumqin sheep. COL3A1, COL1A2, and COL1A1 are related to protein digestion and absorption. COL3A1, COL1A2, and COL1A1 were highly expressed in the somite and notochord in Ujumqin sheep. These results could explain why the morphology of the Uqumqin sheep includes a big tail. RNA velocity and gene expression analyses support the mesenchyme–notochord cell transformation theory in Ujumqin sheep. This can help to explain the role of notochord cells in tail development. This comprehensive single-cell map reveals previously unrecognized signaling pathways that contribute to understanding the mechanism of short-tailed sheep formation.

## Figures and Tables

**Figure 1 vetsci-10-00543-f001:**
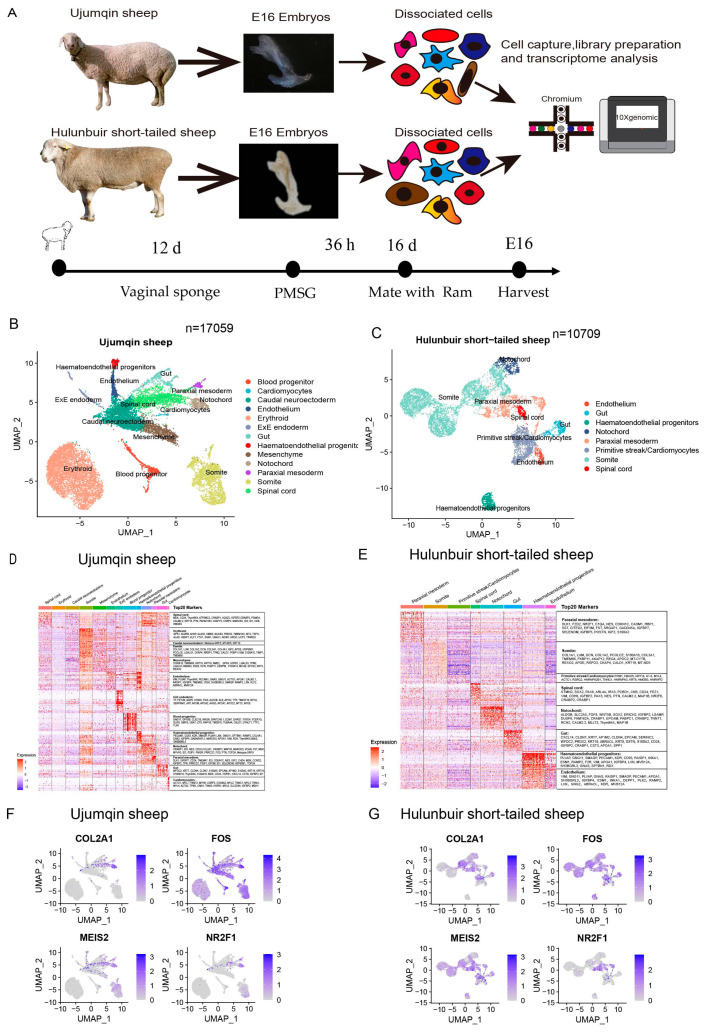
Cell type identification and cluster analysis at E16 in Ujumqin and Hulunbuir sheep. (**A**) Schematic of sheep E16, cell isolation, processing, and capture of single cells using a droplet-based device. (**B**) Anchoring of Ujumqin sheep (*n* = 17,059 viable cells) datasets visualized in UMAP space. (**C**) Anchoring of Hulunbuir sheep (*n* = 10,709 viable cells) datasets visualized in UMAP space. (**D**) Top 20 genes expressed in every cell cluster in Ujumqin sheep. (**E**) Top 20 genes expressed in every cell cluster in Hulunbuir sheep. (**F**,**G**) COL2A1, FOS, MEIS2, and NR2F2 expression level detection in Ujumqin and Hulunbuir sheep (Figure 1 presents our study).

**Figure 2 vetsci-10-00543-f002:**
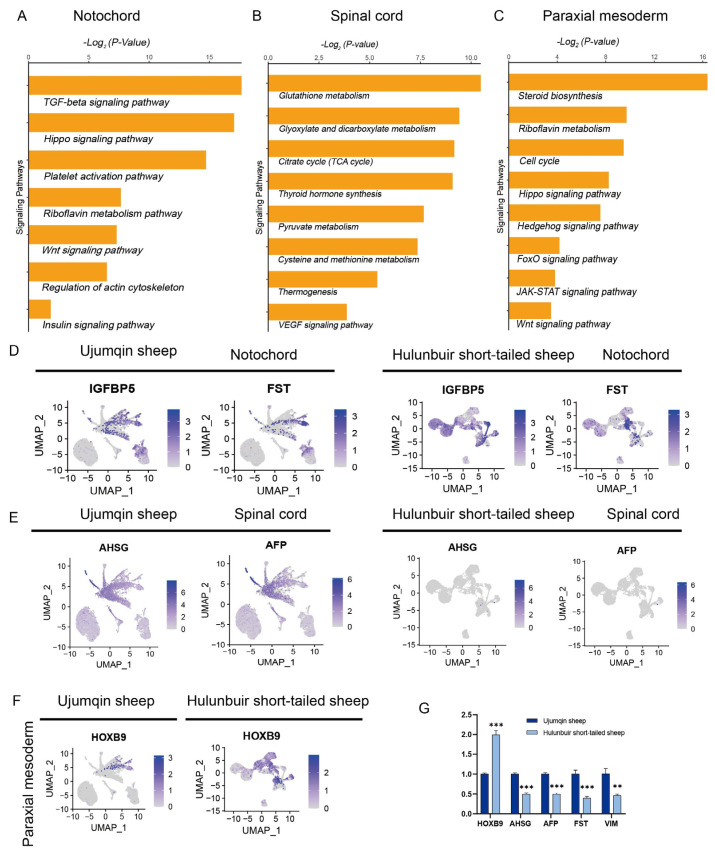
Functional enrichment analysis of E16 in Ujumqin and Hulunbuir sheep. (**A**) KEGG showing signaling pathways activated in the notochord of Ujumqin sheep. These pathways included the TGF-beta signaling pathway, Hippo signaling pathway, platelet activation pathway, riboflavin metabolism pathway, Wnt signaling pathway, regulation of actin cytoskeleton, and insulin signaling pathway. (**B**) KEGG showing signaling pathways activated in the spinal cord in Ujumqin sheep. These pathways included glutathione metabolism, dicarboxylate metabolism, the citrate cycle (TCA cycle), thyroid hormone synthesis, pyruvate metabolism, cysteine and methionine metabolism, thermogenesis, and VEGF signaling pathway. (**C**) KEGG showing signaling pathways activated in the paraxial mesoderm in Ujumqin sheep. These pathways included steroid biosynthesis, riboflavin metabolism, the cell cycle, the Hippo signaling pathway, the Hedgehog signaling pathway, the FoxO signaling pathway, the JAK-STAT signaling pathway, and the Wnt signaling pathway. (**D**) IGFBP5 and FST expression detection. (**E**) AHSG and AFP expression detection. (**F**) HOXB9 expression detection. (**G**) Selected DEGs were confirmed using qRT-PCR (** *p* < 0.01, and *** *p* < 0.001).

**Figure 3 vetsci-10-00543-f003:**
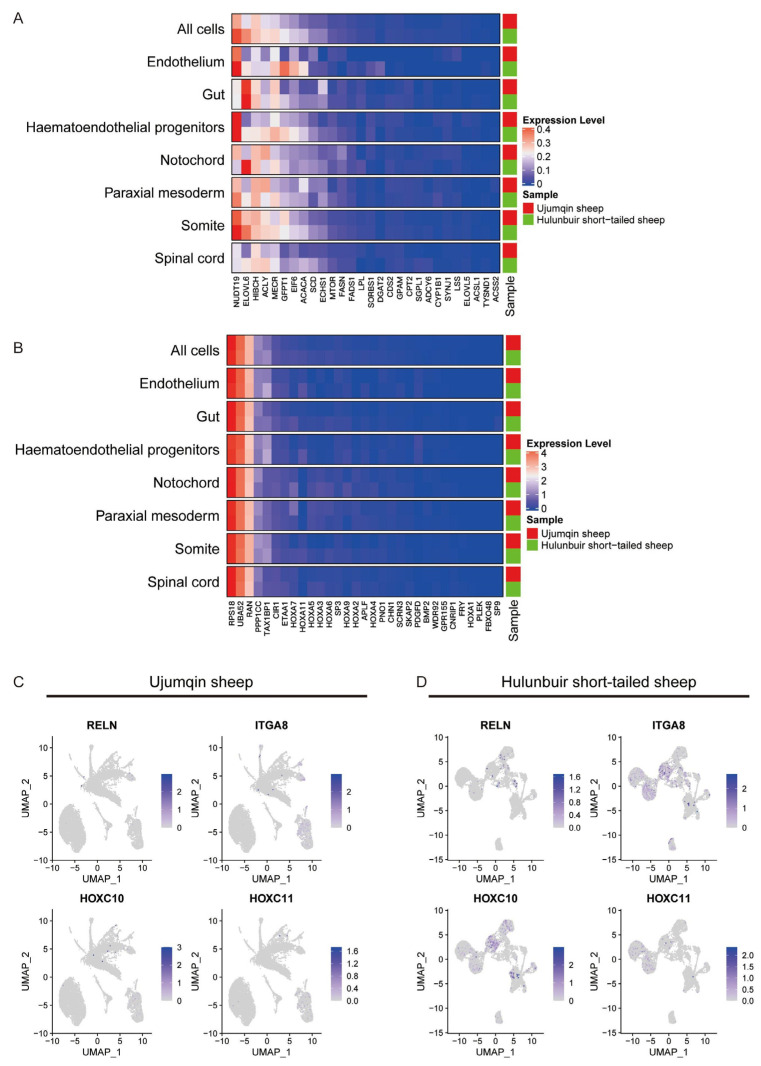
Differential expression gene analysis in every cell cluster at E16 in Ujumqin and Hulunbuir sheep. (**A**) ELOV6, MECR, and ACLY expression analysis in Ujumqin and Hulunbuir sheep. (**B**) PPP1CC (Protein Phosphatase 1 Catalytic Subunit Gamma) expression detection in the notochord, paraxial mesoderm, somite, and spinal cord in Ujumqin and Hulunbuir sheep. (**C**,**D**) REN (extracellular matrix serine protease), ITGA8 (Integrin Submit Alpha8), HOXC10, and HOXC11 expression detection in Ujumqin and Hulunbuir sheep.

**Figure 4 vetsci-10-00543-f004:**
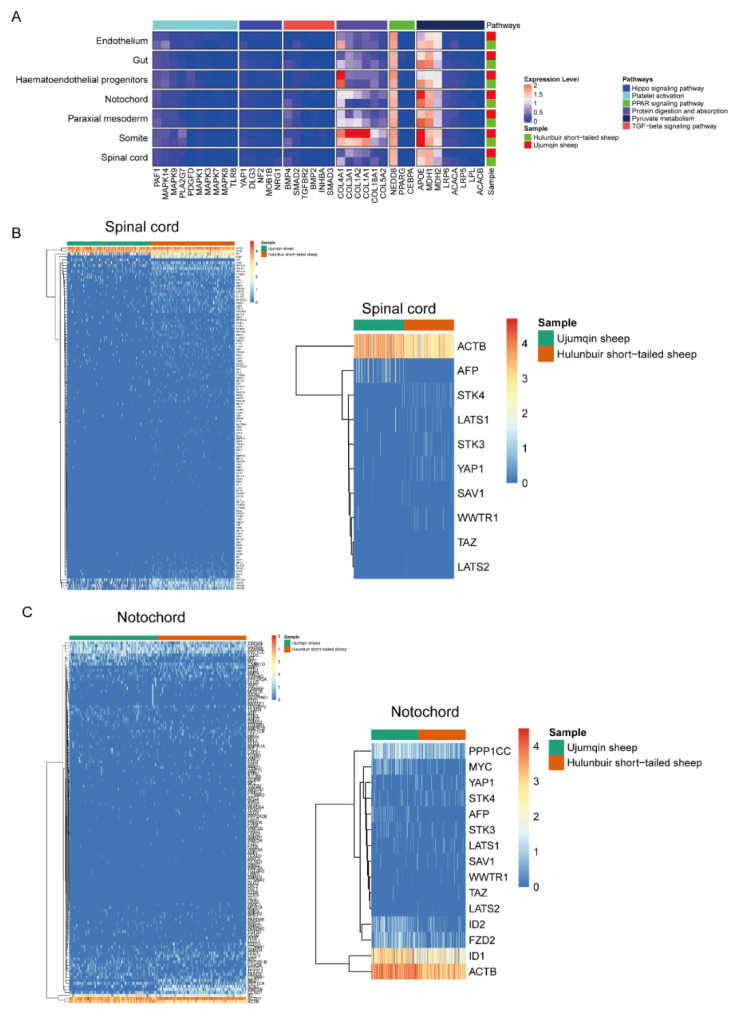
Sc-RNAseq analysis of notochord, paraxial mesoderm, somite, and spinal cord cell characteristics in Hulunbuir and Ujumqin sheep. (**A**) Hippo signaling pathway, TGF-β signaling pathway, COL3A1, COL1A2, COL1A1, and APOE detection in the notochord, paraxial mesoderm, somite, and spinal cord cell characteristics in Hulunbuir and Ujumqin sheep. (**B**) Hippo–YAP signaling pathway detection in the spinal cord. (**C**) Hippo–YAP signaling pathway detection in the notochord.

**Figure 5 vetsci-10-00543-f005:**
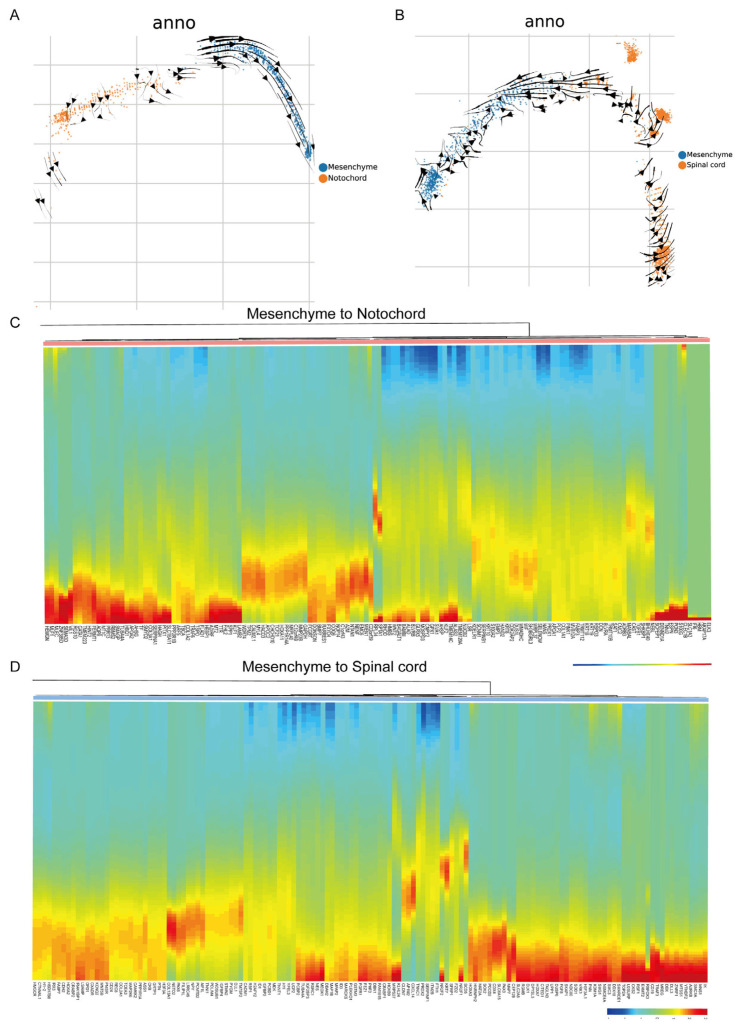
RNA velocity and gene expression analyses support the mesenchyme–notochord cell transformation theory in Ujumqin sheep. (**A**) Pseudo-time analysis of select mesenchyme and notochord cells in Ujumqin sheep. The arrows represent the direction of cell transformation (**B**) Pseudo-time analysis of selected mesenchyme and spinal cord cells in Ujumqin sheep. The arrows represent the direction of cell transformation (**C**) Expression of pseudo-time-dependent transcription factors (TFs) in the notochord. (**D**) Expression of pseudo-time-dependent transcription factors (TFs) in the spinal cord.

**Figure 6 vetsci-10-00543-f006:**
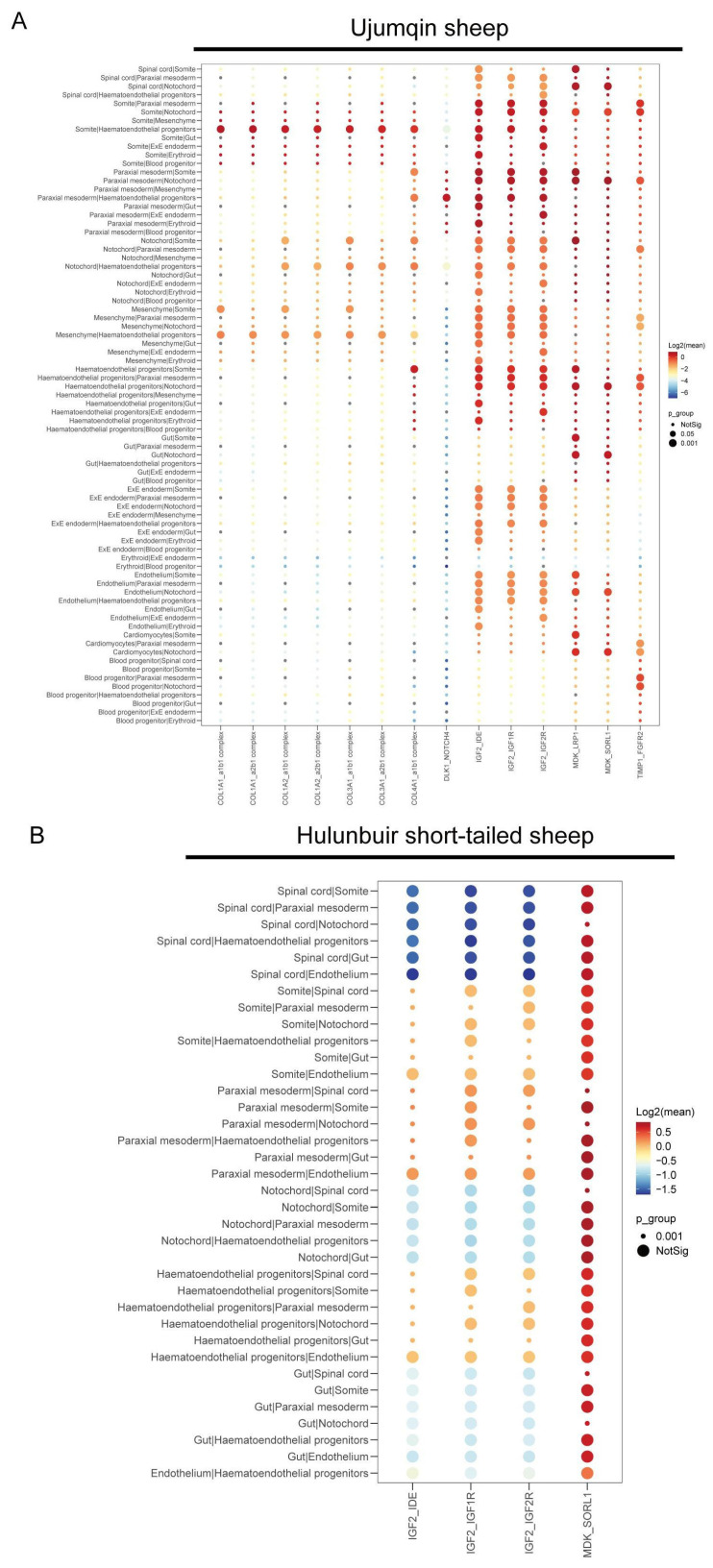
Cell–cell communication analysis in Ujumqin and Hulunbuir sheep. (**A**) Detection of cell–cell communication in Ujumqin sheep. (**B**) Detection of cell–cell communication in Hulunbuir sheep.

## Data Availability

Single-cell RNA sequencing is available from the Gene Expression Omnibus (GEO) database with the accession number GSE185233.

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
