# Peer review of "A Single-Cell Atlas of an Early Mongolian Sheep Embryo"

_vetsci, 2023, doi:10.3390/vetsci10090543_

Round 1

Reviewer 1 Report

VETERINARY SCIENCES - MDPI

Referee’s Evaluation Report

MANUSCRIPT IDENTIFICATION:  VETSCI-2450971 

A single-cell atlas of Mongolia sheep early embryo

 (ORIGINAL ARTICLE)

Comments to Authors/Editor:

The paper of He & colleagues is a comprehensive single-cell transcriptomic characterization study aimed to revel unrecognized signaling pathways in a sheep model during the organogenesis-embryogenesis process in two Mongolian breeds. Therefore, cell populations, marker gene identification and signaling pathway were analyzed. This manuscript falls within the scope of VETERINARY SCIENCES. The manuscript is sufficiently informative for the replication of the study.  In general, the organization of the experiment seems to be well designed, yet the English quality, grammar, and sentence structure must be greatly improved. The Abstract section was written in a careless fashion, with server orthographic errors; this section must be extensively corrected. This manuscript version does not have line numbers, so, is very difficult to suggest corrections; the Editorial Team of VETSCI must help the reviewers during the evaluation process and request to the authors with accurate format specifications. In L4 of the Abstract section the authors must define the meaning of E16; it is fundamental to define this stage for the better understanding of the whole manuscript; why the authors select E16 to perform this study??? The Introduction section is plenty of orthographic mistakes, run-on sentences as well as incomplete sentences; this section must be rewritten. Every sentence must be supported by the proper citation; correct along with the entire manuscript. Are the authors comfortable with the length of the Introduction section???, it is extremely large and in its actual format is extremely confusing; please shorten this section. While the objectives of the study were stated, no working hypothesis of the study was proposed; this is a must. Please include the contribution of sheep to the national livestock sector; why do the authors use sheep as a research model instead buffaloes, cows, or goats??? Are sheep, in inner Mongolia, China, important from a productive, economic, or social standpoint??? Some information included in the Introduction section can be moved to the Discussion section. In the Material & Methods section, I strongly recommend including a figure with the actual experimental protocol across time (i.e. a timeline of actions); this is a must. Besides, the authors must indicate how the experimental samples were managed during the experimental period, including more information regarding the two sheep breed analyzed (i.e., live weight, body condition, production system). Both, in the Abstract and M&M sections, the authors must include where the experiment was carried out (NL, WL), including the months when the samples were collected, and define if the used breeds depict a seasonal or continuous reproduction; what season of the year were the sampling performed. Please state, if is the case, whether the experimental samples were collected at the beginning, the middle, or the last stage of the breeding season.  As mentioned, every statement must include at least one reference; it is real nonsense not to include the required bibliographical sources in any scientific report; correct accordingly along with the manuscript. In the M&M section, the authors must define if the samples were collected from homogeneous experimental units??? That is, the age, body weight, body condition score, lactation number; please carefully explain the physiological nature of the collected samples. In general, the materials, standards, and methods used are relevant and in accordance with the objectives of the study. Also, all the sampling techniques, the reported methodology, and consider techniques used in the experiment are detailed and accurate, while in accordance with the objectives of the study. Again, the English quality must be improved also in this section. In this section, the authors must be very careful to clearly define the number of replicates analyzed per experimental group. Again, both the M&M as well as the Results section include several uncomplete, non-sense sentence; please correct accordingly. The authors are including citations in the Results section; the information must be removed towards the Discussion section; it is quite confused. In general, in the Results section the novelty value of the results is reasonable, yet the authors must include some kind of quantitative information involving the P-values of the observed results regarding the response variables.  With respect to the Discussion section, I do strongly suggest initiating this section including the working hypothesis of the study. Authors must define if, with the obtained results, such a hypothesis is rejected or non-rejected. For this reason, the authors must include the working hypothesis prior to the objectives in the Introduction section. In addition, the authors must follow the same order in this section according to that proposed in the Results section; the authors must homogenate the presentation of the Results and Discussion.  The authors must link, in a logical fashion, their main findings along with the discussion section, to compare & discuss and, thereafter, be able to propose some physiologic explanations at genomic level (i.e. gene level), for such specific outcomes, considering previous similar studies from the scientific literature. In general, the authors made an accurate interpretation of the main findings.  The Discussion section is quite extended; the authors must focus their main findings and confront them with respect to the scientific literature in a logical and focused fashion.  The main outcomes of the study were neither clearly nor soundly discussed. With respect to the Conclusion section, the authors must highlight the main findings of the study and the possible use of the study outcomes upon sheep’ production; conclusions must be aligned with the working hypothesis. The list of references cited in the manuscript is proper (51), while actualized. The authors also include very interesting supplementary information.  This is an interesting study, with a large set of response variables. Yet, on one side the authors must improve both the English language quality as well as the clarity and logical arrangement of the observed results, especially in the Results and Discussion sections. Sorry about this situation but it is necessary to ensure that the paper is readable.  The authors must increase the readability and the scientific writing and merit of the manuscript.  All the commented issues and requests should be clearly addressed by the authors; at this point, and based on the above comments, my pronouncement is that this manuscript cannot be accepted in its actual format.  It requires extensive editions and corrections.

Must be improved....

Author Response

Dear Reviewer

We thank  reviewer for  enthusiasm and  many insightful comments on our manuscript (2450971). We have comprehensively addressed all of the comments from Reviewer. We sincerely hope that the revised and improved manuscript is now suitable for publication in VETERINARY SCIENCES . Below please find our detailed point-by-point response to the Reviewers.

Reviewer1

Comments to Authors/Editor:

The paper of He & colleagues is a comprehensive single-cell transcriptomic characterization study aimed to revel unrecognized signaling pathways in a sheep model during the organogenesis-embryogenesis process in two Mongolian breeds. Therefore, cell populations, marker gene identification and signaling pathway were analyzed. This manuscript falls within the scope of VETERINARY SCIENCES. The manuscript is sufficiently informative for the replication of the study. 

  1. In general, the organization of the experiment seems to be well designed, yet the English quality, grammar, and sentence structure must be greatly improved.

Response: We thank the reviewer for his/her suggestion. We have improved the English quality, grammar, and sentence structure of the manuscript with help of many experts and MDPI  language editing services.

  1. The Abstract section was written in a careless fashion, with server orthographic errors; this section must be extensively corrected.

Response: We thank the reviewer for his/her suggestion. We have corrected it in revised manuscript.

  1. This manuscript version does not have line numbers, so, is very difficult to suggest corrections; the Editorial Team of VETSCI must help the reviewers during the evaluation process and request to the authors with accurate format specifications

Response: We thank the reviewer for his/her suggestion.We have added line numbers.

  1. In L4 of the Abstract section the authors must define the meaning of E16; it is fundamental to define this stage for the better understanding of the whole manuscript; why the authors select E16 to perform this study???

Response: We thank the reviewer for his/her suggestion. TBXT mutations are related to tailless or short-tail phenotypes in vertebrates. TBXT is the highest expression in E16.

  1. The Introduction section is plenty of orthographic mistakes, run-on sentences as well as incomplete sentences; this section must be rewritten. Every sentence must be supported by the proper citation; correct along with the entire manuscript. Are the authors comfortable with the length of the Introduction section???, it is extremely large and in its actual format is extremely confusing; please shorten this section. While the objectives of the study were stated, no working hypothesis of the study was proposed; this is a must. Please include the contribution of sheep to the national livestock sector; why do the authors use sheep as a research model instead buffaloes, cows, or goats??? Are sheep, in inner Mongolia, China, important from a productive, economic, or social standpoint??? Some information included in the Introduction section can be moved to the Discussion section.

Response: We thank the reviewer for his/her suggestion.We have corrected these.We found sheep short-tailed phenotype.so, we used sheep as a research model. Sheep are very important  in inner Mongolia, China from a productive, economic standpoint.

  1. In the Material & Methods section, I strongly recommend including a figure with the actual experimental protocol across time (i.e. a timeline of actions); this is a must. Besides, the authors must indicate how the experimental samples were managed during the experimental period, including more information regarding the two sheep breed analyzed (i.e., live weight, body condition, production system). Both, in the Abstract and M&M sections, the authors must include where the experiment was carried out (NL, WL), including the months when the samples were collected, and define if the used breeds depict a seasonal or continuous reproduction; what season of the year were the sampling performed.

Response: We thank the reviewer for his/her suggestion.We have corrected these.

  1. Please state, if is the case, whether the experimental samples were collected at the beginning, the middle, or the last stage of the breeding season.  As mentioned, every statement must include at least one reference; it is real nonsense not to include the required bibliographical sources in any scientific report; correct accordingly along with the manuscript. In the M&M section, the authors must define if the samples were collected from homogeneous experimental units??? That is, the age, body weight, body condition score, lactation number; please carefully explain the physiological nature of the collected samples. In general, the materials, standards, and methods used are relevant and in accordance with the objectives of the study. Also, all the sampling techniques, the reported methodology, and consider techniques used in the experiment are detailed and accurate, while in accordance with the objectives of the study. Again, the English quality must be improved also in this section. In this section, the authors must be very careful to clearly define the number of replicates analyzed per experimental group. Again, both the M&M as well as the Results section include several uncomplete, non-sense sentence; please correct accordingly. The authors are including citations in the Results section; the information must be removed towards the Discussion section; it is quite confused. In general, in the Results section the novelty value of the results is reasonable, yet the authors must include some kind of quantitative information involving the P-values of the observed results regarding the response variables.  With respect to the Discussion section, I do strongly suggest initiating this section including the working hypothesis of the study. Authors must define if, with the obtained results, such a hypothesis is rejected or non-rejected. For this reason, the authors must include the working hypothesis prior to the objectives in the Introduction section. In addition, the authors must follow the same order in this section according to that proposed in the Results section; the authors must homogenate the presentation of the Results and Discussion.  The authors must link, in a logical fashion, their main findings along with the discussion section, to compare & discuss and, thereafter, be able to propose some physiologic explanations at genomic level (i.e. gene level), for such specific outcomes, considering previous similar studies from the scientific literature. In general, the authors made an accurate interpretation of the main findings.  The Discussion section is quite extended; the authors must focus their main findings and confront them with respect to the scientific literature in a logical and focused fashion.  The main outcomes of the study were neither clearly nor soundly discussed. With respect to the Conclusion section, the authors must highlight the main findings of the study and the possible use of the study outcomes upon sheep’ production; conclusions must be aligned with the working hypothesis. The list of references cited in the manuscript is proper (51), while actualized. The authors also include very interesting supplementary information.  

Response: We thank the reviewer for his/her suggestion.We have corrected these.

  1. This is an interesting study, with a large set of response variables. Yet, on one side the authors must improve both the English language quality as well as the clarity and logical arrangement of the observed results, especially in the Results and Discussion sections. Sorry about this situation but it is necessary to ensure that the paper is readable.  The authors must increase the readability and the scientific writing and merit of the manuscript.  All the commented issues and requests should be clearly addressed by the authors; at this point, and based on the above comments, my pronouncement is that this manuscript cannot be accepted in its actual format.  It requires extensive editions and corrections.

Response: We thank the reviewer for his/her suggestion. We have corrected these.

Reviewer 2 Report

Dear authors

The subject is quite interesting, the methodology also; yet the paper lacks in the way of presentation and the writing of several parts. Many references are missing and the english are really hard to follow. I suggest that you re-write the whole manuscript, and it has to be checked by a native speaker before resubmitting.

There are errors in verbs, in punctuation, some sentences lack of meaning.

Author Response

Dear Reviewer

We thank  reviewer for  enthusiasm and  many insightful comments on our manuscript (2450971). We have comprehensively addressed all of the comments from Reviewer. We sincerely hope that the revised and improved manuscript is now suitable for publication in VETERINARY SCIENCES . Below please find our detailed point-by-point response to the Reviewers.

Reviewer2

The subject is quite interesting, the methodology also; yet the paper lacks in the way of presentation and the writing of several parts. Many references are missing and the english are really hard to follow. I suggest that you re-write the whole manuscript, and it has to be checked by a native speaker before resubmitting.

Response: We thank the reviewer for his/her suggestion. We have improved the English quality, grammar, and sentence structure of the manuscript with help of many experts and MDPI  language editing services.

Round 2

Reviewer 1 Report

The R1 version of the manuscript was improved...

Author Response

Dear  reviewer

We thank  reviewer for their enthusiasm and  many insightful comments on our manuscript (2450971). We have comprehensively addressed all of the comments from Reviewer. We sincerely hope that the revised and improved manuscript is now suitable for publication in VETERINARY SCIENCES . 

Round 3

Reviewer 1 Report

Referee’s Evaluation Report

MANUSCRIPT IDENTIFICATION:  VETSCI-2450971 – R2

A single-cell atlas of Mongolia sheep early embryo

 (ORIGINAL ARTICLE)

Comments to Authors/Editor:

This is the R2 version of the paper of He & colleagues. The study is a comprehensive single-cell transcriptomic characterization aimed to revel unrecognized signaling pathways in a sheep model during the organogenesis-embryogenesis process in two Mongolian breeds. Therefore, cell populations, marker gene identification and signaling pathway were analyzed. In this R2 version, the English quality, grammar, and sentence structure were certainly improved. The authors mentioned why they selected the E16 stage to perform this study; however, again, the authors never define what E16 means; please include such information in the final version. The Introduction section was improved. While the objectives of the study were stated, again no working hypothesis of the study was proposed as previously requested; please include the working hypothesis in the final version.  Again, what sense does it make for us as reviewers to make suggestions to improve the scientific rigor of the study, if the authors do not take our suggestions into account? In this R2 version the authors did explain why they used sheep as a research model instead other domestic animal. Both the Material & Methods, and the Results sections were also improved; citations were eliminated in the Results section. As mentioned in the R1 review, the materials, standards, and methods used are relevant and in accordance with the objectives of the study. Also, all the sampling techniques, the reported methodology, and consider techniques used in the experiment are detailed and accurate, while in accordance with the objectives of the study. The English quality was also improved in this section. With respect to the Discussion section was also improved; yet, despite it was previously suggested to initiate this section including the working hypothesis of the study, the authors did not consider such suggestion; please, the authors must define if, with the obtained results, such a hypothesis is rejected or non-rejected. For this reason, we also proposed to the authors the inclusion of the working hypothesis prior to the objectives in the Introduction section; this section was also improved. The Conclusion section is correct, while the list of references cited in the manuscript is proper and actualized. The authors also include very interesting supplementary information. The authors improved both the English language quality as well as the clarity and logical arrangement of the observed results, especially in the Results and Discussion sections. At this point, and based on the above comments, my pronouncement is that this manuscript CAN BE ACCEPTED for publication once they include the requested information.

Accept after minor corrections.

Author Response

Dear editor and reviewer

   Please see the attachment,thanks!
